# ZipCache: Accurate and Efficient KV Cache Quantization with Salient Token Identification

**Yefei He[1]    Luoming Zhang[1]    Weijia Wu[2]    Jing Liu[3]**
**Hong Zhou[1]***    **Bohan Zhuang[1,3]***

[1]Zhejiang University, China
[2]National University of Singapore, Singapore
[3]ZIP Lab, Monash University, Australia

## Abstract

KV cache stores key and value states from previous tokens to avoid re-computation, yet it demands substantial storage space, especially for long sequences. Adaptive KV cache compression seeks to discern the saliency of tokens, preserving vital information while aggressively compressing those of less importance. However, previous methods of this approach exhibit significant performance degradation at high compression ratios due to inaccuracies in identifying salient tokens. Additionally, the compression process introduces excessive overhead, substantially increasing memory burdens and the generation latency. In this paper, we present ZipCache, an accurate and efficient KV cache quantization method for large language models (LLMs). First, we construct a strong baseline for quantizing KV cache. Through the proposed channel-separable tokenwise quantization scheme, the memory overhead of quantization parameters are substantially reduced compared to fine-grained groupwise quantization. To enhance the compression ratio, we propose normalized attention score as an effective metric for identifying salient tokens by considering the lower triangle characteristics of the attention matrix. The quantization bit-width for each token is then adaptively assigned based on their saliency. Moreover, we develop an efficient approximation method that decouples the saliency metric from full attention scores, enabling compatibility with fast attention implementations like FlashAttention. Extensive experiments demonstrate that ZipCache achieves superior compression ratios, fast generation speed and minimal performance losses compared with previous KV cache compression methods. For instance, when evaluating Mistral-7B model on GSM8k dataset, ZipCache is capable of compressing the KV cache by $4.98\times$, with only a $0.38\%$ drop in accuracy. In terms of efficiency, ZipCache also showcases a $37.3\%$ reduction in prefill-phase latency, a $56.9\%$ reduction in decoding-phase latency, and a $19.8\%$ reduction in GPU memory usage when evaluating LLaMA3-8B model with a input length of 4096. Code is available at https://github.com/ThisisBillhe/ZipCache/.

## 1   Introduction

LLMs with the next-token-prediction scheme have achieved remarkable advancements in various text-related tasks, such as language understanding [13, 34, 10], content creation [1, 5, 36], coding [3, 29, 42] and mathematics [33, 23, 35]. In this generation scheme, the forthcoming token interacts with all previous tokens via the attention mechanism [38], where the query, key and value states will be

---

*Corresponding author. Email: `zhouhong_zju@zju.edu.cn`, `bohan.zhuang@gmail.com`

38th Conference on Neural Information Processing Systems (NeurIPS 2024).

calculated for each token. As the past tokens will not be altered, previously computed key and value states can be stored as KV cache to prevent re-computations, significantly improving the generation speed. However, as the batch size and the input context length grows, the stored KV cache emerges as a new memory bottleneck for LLMs. For example, when serving a 175B-parameter LLM [1] with a batch size of 64 and a context length of 4096, the KV cache can occupy 1.2TB of memory space, while the model weights only require 350GB. Meanwhile, the size of KV cache will continue to increase as decoding progresses. Therefore, the compression of KV cache is crucial for the efficient deployment of LLMs.

Recent compression methods for KV cache can be broadly categorized into two types. The first type of methods compresses the KV cache uniformly, without considering the significance of individual tokens. To preserve performance, these methods often rely on either high-precision quantization [21] or maintaining recent tokens in full-precision [32], which undoubtedly compromise the compression ratio. Additionally, if salient tokens are not among the most recent ones, such as in information retrieval tasks, it may result in degraded performance. The other type of methods [46, 43, 16] compress KV cache adaptively by identifying salient tokens and compresses them separately. This approach aligns with the observation that a minority of tokens contribute the majority of attention scores [41], potentially achieving higher compression ratios than non-adaptive methods. However, current adaptive KV cache compression methods [46, 43] use accumulated attention scores as a metric of token saliency, which is insufficient in two aspects. First, accumulated attention scores is **inaccurate** in identifying important tokens. Due to the presence of attention masks, the attention matrix is a lower triangular matrix. Earlier tokens tend to have larger softmax attention values and more attention scores to be accumulated, as illustrated in Figure 3. Under this metric, the saliency of the most recent tokens can never surpass that of the first token, thereby introducing a bias in determining token saliency. Additionally, to obtain accumulated attention scores, full attention matrices must be explicitly computed and stored, which can be **inefficient** for serving LLMs. Given an input context length of $l$, fast attention implementations such as FlashAttention [8, 7] only require $O(l)$ memory by computing attention output in blocks without retaining complete attention matrices. By contrast, storing full attention matrices requires $O(l^2)$ memory, and the large number of memory accesses significantly slows down the inference speed, as depicted in Figure 4.

To address these challenges, we introduce ZipCache, an efficient KV cache compression method that attains exceptionally high compression ratios by accurate salient token identification. Figure 1 presents an overview of latency-accuracy comparisons among ZipCache and diverse KV cache compression methods. We start by designing an efficient quantization baseline for compressing the KV cache. To preserve performance, predecessor methods [32, 21] employ fine-grained groupwise quantization, which involves independent quantization for a small channel group within each token. However, this method necessitates storing extensive quantization parameters and results in significant memory overhead. By contrast, we introduce a channel-separable quantization scheme that decouples the quantization along channel and token dimensions. This method significantly reduces the quantization overhead without compromising performance. To accurately recognize salient tokens, we introduce a new token saliency metric based on normalized attention scores, which alleviates the bias towards earlier tokens that accumulate more values. All tokens, without exception, will be quantized to the target bit-width based on their estimated saliency, boosting the overall compression ratio. Moreover, to

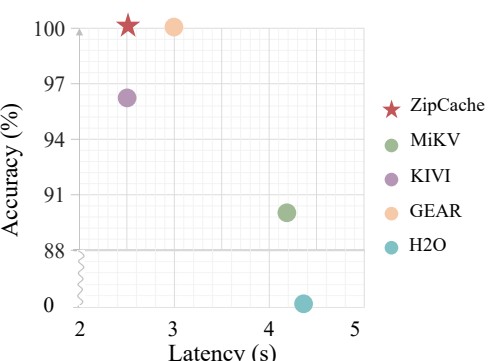

Figure 1: Accuracy and efficiency comparisons across various KV cache compression methods. Data is collected with LLaMA3-8B model on Line Retrieval dataset. Among these methods, ZipCache achieves the highest accuracy, generation speed and compression ratio. Details can be found in the supplementary material.

ease integration with fast attention implementations, we introduce an efficient approximation of the token saliency metric. This approximation only relies on computing and storing attention scores from a few number of tokens, which we refer to as probe tokens. An effective probe token selection strategy is then introduced to minimize performance loss. As a result, the majority of tokens can benefit from fast attention implementations, significantly enhancing the generation speed.

In summary, our contributions are as follows:

- We establish an efficient channel-separable quantization scheme for KV cache, which significantly reduces the overhead of quantization parameters without compromising performance compared to fine-grained groupwise quantization approach.
- We propose an accurate metric for assessing token saliency based on normalized attention scores. All tokens are adaptively quantized according to their assessed saliency, thereby improving the overall compression ratio.
- We further develop an efficient approximation method for the token saliency metric that integrates seamlessly with fast attention implementations, enhancing generation speed.
- By integrating these three techniques, we present ZipCache, an accurate and efficient framework for KV cache compression. Extensive experiments demonstrate that ZipCache reaches a new state-of-the-art performance for KV cache compression in terms of compression ratio, accuracy and generation efficiency.

## 2 Related Work

### 2.1 Model Quantization

Quantization is a prevalent technique for compressing deep neural networks by representing model weights and activations with lower numerical bit-widths. This technique can be categorized into two primary approaches based on the necessity of fine-tuning: post-training quantization (PTQ) [26, 17, 14] and quantization-aware training (QAT) [28, 31]. For large language models (LLMs), where fine-tuning can be data- and computation-intensive, PTQ is often the preferred method [40, 11, 45, 27]. In this paper, we also quantize KV cache in a post-training manner. For both approaches, quantization can be implemented at various levels of granularity, including channelwise, tokenwise, and groupwise approach. Typically, a finer quantization granularity involves the independent quantization of smaller parameter groups, which often results in improved performance albeit at the cost of more quantization parameters and increased memory overhead. In the context of LLMs, fine-grained quantization is frequently utilized due to the presence of outliers [22, 45]. However, for KV cache compression, this will greatly reduce the overall compression ratio.

Mixed precision quantization [39, 44, 12, 2] allocates varying bit-widths to distinct parts of a model or tensor, enabling a more compact compression. This approach originates from the observation that model components exhibit differing sensitivities to quantization. Consequently, components with low sensitivity can utilize reduced bit-widths without impairing performance. For LLMs, previous studies [46, 43, 30, 18] have shown significant disparities in the importance of tokens, indicating that heavy compression of non-critical tokens has minimal impact on overall performance. This insight highlights the applicability of mixed precision quantization for compressing the KV cache.

### 2.2 KV Cache Compression

While KV cache effectively prevents re-computation and significantly enhances generation speed, its memory footprint is notably substantial with long-context input. To alleviate this, many efforts have been made to reduce the KV cache size. Based on the compression method, these methods can be categorized into two groups: token dropping [46, 16, 30] and KV cache quantization [43, 21, 32]. The former identifies and drops unimportant tokens in the KV cache. For example, H2O [46] only maintain 20% heavy-hitted tokens and 20% recent tokens while evicting the rest. However, discarding tokens permanently erases their information, which proves to be suboptimal for tasks such as retrieval [43]. Conversely, the latter category employs quantization on the cached key and value states, and mixed precision quantization can further be applied once token importance is identified [43]. To tackle the outliers present in the KV cache, these methods extract the outlier as full precision [21] or use finer-grained quantization scheme [32], which increases the quantization overhead. In this study, we propose an efficient channel-separable quantization scheme with reduced quantization overhead and strong performance. Additionally, both categories of methods commonly adopt accumulated attention scores as the metric for token importance [46, 43]. However, we observe that this criterion is inaccurate and can result in significant performance deterioration at low bit-widths. In contrast, we achieve superior compression performance by utilizing a more accurate metric for identifying salient tokens.

## 3 Preliminary

### 3.1 Attention Block in LLMs

Given an input prompt, the generation process of LLMs can be broadly categorized into two distinct phases: the prefill phase, which computes and stores the KV cache for input tokens, and the decoding phase, where new tokens are generated through a next-token-prediction scheme. Given input data $\mathbf{X}$ and an attention block with its weight matrices $\mathbf{W}_Q$, $\mathbf{W}_K$ and $\mathbf{W}_V$, the prefill phase can be formulated as:

$$\mathbf{Q} = \mathbf{X}\mathbf{W}_Q, \quad \mathbf{K} = \mathbf{X}\mathbf{W}_K, \quad \mathbf{V} = \mathbf{X}\mathbf{W}_V, \tag{1}$$

$$\mathbf{A} = \mathrm{Softmax}\left(\frac{\mathbf{Q}\mathbf{K}^T}{\sqrt{d_k}}\right), \quad \mathbf{O} = \mathbf{A}\mathbf{V}. \tag{2}$$

Here, $d_k$ is the dimension of the key, and $\mathbf{A}$ refers to the attention scores. $\mathbf{K}$ and $\mathbf{V}$ will be stored as KV cache. For clarity, we have omitted the output projection.

For the decoding phase, given $\mathbf{x}$ as the embedding vector of the current token, the query $\mathbf{q}$ becomes a vector and the KV cache matrices will be updated as follow:

$$\mathbf{q} = \mathbf{x}\mathbf{W}_Q, \quad \mathbf{K} = \mathrm{Concat}(\mathbf{K}, \mathbf{x}\mathbf{W}_K), \quad \mathbf{V} = \mathrm{Concat}(\mathbf{V}, \mathbf{x}\mathbf{W}_V). \tag{3}$$

The attention output are then computed as follows:

$$\mathbf{a} = \mathrm{Softmax}\left(\frac{\mathbf{q}\mathbf{K}^T}{\sqrt{d_k}}\right), \quad \mathbf{o} = \mathbf{a}\mathbf{V}. \tag{4}$$

To ensure clarity and consistency, we introduce notation to define the hyper-parameters used in the paper. Specifically, we denote the batch size as $b$, the number of attention heads as $h$, the sequence length as $l$, and the head dimension as $d$.

### 3.2 Model Quantization

Uniform quantization is adopted in our study and all experiments. Given a floating-point vector $\mathbf{x}$, it can be uniformly quantized to $k$-bit as follows:

$$\hat{\mathbf{x}} = \mathcal{Q}_U(\mathbf{x}, k) = (\mathrm{clip}(\lfloor\frac{\mathbf{x}}{s}\rceil + z, 0, 2^k - 1) - z) \cdot s. \tag{5}$$

Here, $\lfloor\cdot\rceil$ denotes the round operation, $s = \frac{\max(\mathbf{x}) - \min(\mathbf{x})}{2^k - 1}$ and $z = -\lfloor\frac{\min(\mathbf{x})}{s}\rceil$ are quantization parameters. It should be noted that the quantization parameters are stored in full-precision, which can lead to significant overhead if the quantization is fine-grained.

## 4 Method

### 4.1 A Strong Baseline for KV Cache Quantization

Tokenwise quantization, as depicted in Figure 2(b) is prevalent in quantizing large language models (LLMs) due to the distinct representations of individual tokens. However, it has been widely observed, as illustrated in Figure 2(a), that outliers emerge within the channel dimensions of key and value matrices [43, 32], posing challenges for tokenwise quantization. To address this, recent work [32] resorts to groupwise quantization, where outlier channels are processed in distinct groups, as illustrated in Figure 2(c). However, this fine-grained quantization approach introduces excessive memory overhead, thereby significantly impacting the compression ratio. For instance, considering $\mathbf{X} \in \mathbb{R}^{b \times h \times l \times d}$ as the data to be quantized and a group size of $n$, tokenwise quantization only results in $2bl$ quantization parameters, while groupwise quantization would yield $\frac{2bhld}{n}$ quantization parameters. Since these parameters are usually stored in full precision, this overhead would constitute a substantial portion of the storage cost for quantized data.

Motivated by depthwise separable convolution [19], we introduce an efficient channel-separable tokenwise quantization scheme, which disentangles the channel and token dimensions. As shown in

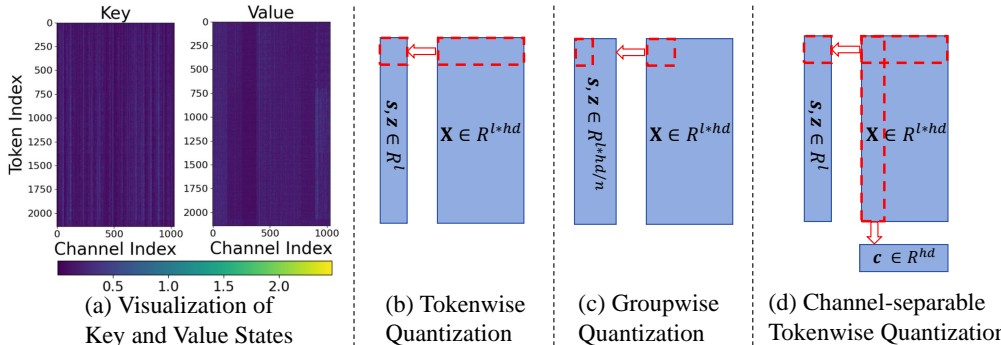

|  | Key | Value |  |  |  |
|---|---|---|---|---|---|
| (a) Visualization of Key and Value States | | | (b) Tokenwise Quantization | (c) Groupwise Quantization | (d) Channel-separable Tokenwise Quantization |

Figure 2: Visualization and different quantization granularities for key and value states. Here, we omit the batch dimension for simplicity. For keys, channel outliers emerge, yet token representations exhibit minimal differences. For values, both channel outliers and distinct token representations exist.

Figure 2(d), our approach initiates by normalizing each channel of data $\mathbf{X}$ with a scaling factor $\mathbf{c}$. For the $i$-th channel in $\mathbf{X}$, the normalization process can be formulated as:

$$\mathbf{X}_i = \frac{\mathbf{X}_i}{\mathbf{c}_i}, \text{ where } \mathbf{c}_i = \sqrt{\max(|\mathbf{X}_i|)}. \tag{6}$$

After normalization, each channel is scaled to a closed magnitude, mitigating the influence of outliers during tokenwise quantization. Subsequently, tokenwise quantization can be reliably applied and the scales $\mathbf{c}$ are multiplied back to restore the magnitude of each channel. The process of channel-separable tokenwise quantization is summarized in the supplementary material. Within this quantization scheme, the total number of quantization parameters amounts to $hd + 2bl$, representing a notable reduction compared to groupwise quantization, while effectively balancing the outlier channels and the representation of each token.

Table 1: Performance comparisons of different quantization granularities for KV cache. The KV cache is quantized to 4-bit and the compression ratio is calculated with $b = 8$, $hd = l = 4096$ and $n = 32$. Data is collected with LLaMA3-8B model on GSM8k dataset.

| Key Cache Quantization Granularity | Value Cache Quantization Granularity | Quantization Parameters | Compression Ratio | Acc.(%) |
|---|---|---|---|---|
| / | / | 0 | 1× | 55.88 |
| Groupwise | Groupwise | $4bhld/n$ | 3.2× | 54.51 |
| Tokenwise | Tokenwise | $4bl$ | 3.99× | 49.81 |
| Channelwise | Tokenwise | $2hd + 2bl$ | 4.00× | 52.77 |
| Channelwise | Channel-separable Tokenwise | $3hd + 2bl$ | 4.00× | **54.74** |

As referred to Figure 2(a), since the differences in token representations are small in key cache, we employ channelwise quantization for the key cache to further reduce overhead and employ channel-separable tokenwise quantization for the value cache. As depicted in Table 1, this configuration yields superior performance with reduced quantization overhead compared with groupwise quantization, thereby establishing a robust baseline for KV cache quantization.

## 4.2 Accurate Salient Token Identification

Adaptive KV cache compression [46, 43, 16] aims to discern the saliency of each token, keeping the information of salient tokens while evicting or aggressively compressing the rest, to achieve a higher compression ratio. These salient tokens, also referred to as "Heavy Hitters" [46], are often identified based on accumulated attention scores. Given attention score matrix $\mathbf{A} \in \mathbb{R}^{l \times l}$, the saliency of token $i$ is estimated by:

$$p_i = \sum_{k=1}^{l} \mathbf{A}_{k,i}. \tag{7}$$

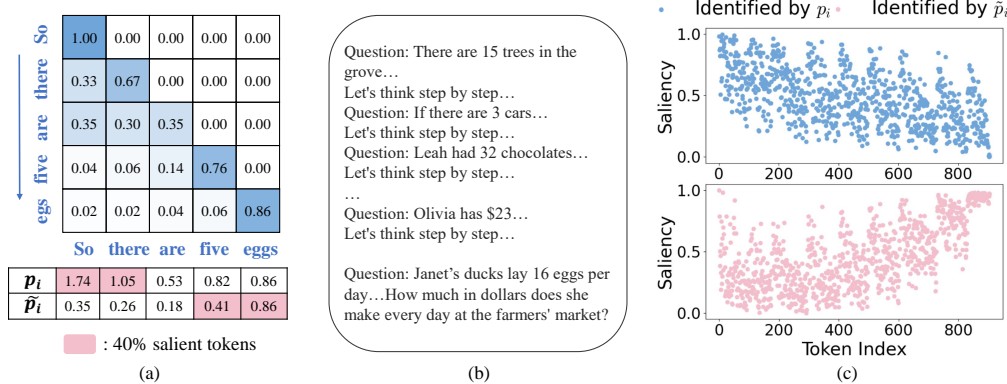

Figure 3: (a) A toy example to illustrate accumulated attention scores and normalized attention scores. Initial tokens have larger attention scores and more values to be accumulated. (b) A sample from GSM8k dataset with chain-of-thoughts (CoT) prompting. (c) The probability of each token being selected as a salient token, measured by both accumulated and normalized attention scores. Tokens correspond to the final question are identified as low saliency by accumulated attention scores.

Tokens with large saliency values are then considered salient tokens. However, this approach has inherent limitations due to the lower triangular nature of the attention score matrix, as illustrated in Figure 3(a). There are two primary issues. **Firstly**, earlier tokens benefit from having more values accumulated since the elements above the diagonal are all zero. For instance, in a sequence of length $l$, the initial token accumulates $l$ positive values, whereas the final token only accumulates one. **Secondly**, Softmax function converts real numbers into probabilities, so that the earlier rows of the attention matrix tending to have higher values, as fewer numbers are involved in the Softmax calculation. Consequently, the accumulated attention score of the final token will always be smaller than that of the first, which exceeds 1. To address this, previous works, such as H2O [46], always maintain recent caches in full precision. Nevertheless, this solution is suboptimal since recent tokens are not necessarily the most significant ones.

To enhance the evaluation of each token's saliency, we introduce an accurate token saliency metric based on normalized attention scores $\tilde{p}_i$:

$$\tilde{p}_i = \frac{\sum_{k=1}^{l} \mathbf{A}_{k,i}}{\text{nnz}(\mathbf{A}_{:,i})} \tag{8}$$

Here, $\text{nnz}(\mathbf{A}_{:,i})$ denotes the number of non-zero elements in the $i$-th column of $\mathbf{A}$. As evidenced in Figure 3(a), normalizing the accumulated attention scores mitigates the influence of excessively large values in the initial rows of the attention score matrix, thereby delivering a more precise assessment. To validate the efficacy of our new metric, we input a sample from GSM8k dataset with chain-of-thoughts (CoT) prompting to the LLaMA3-8B model and identify saliency of each token by Eq. 7 and Eq. 8, respectively. As depicted in Figure 3(b) and (c), the salient tokens are at the end of the prompt, which correspond to the question for LLM to answer. However, these tokens are identified as low saliency by accumulated attention scores. Under the KV cache compression framework, these tokens would either be discarded or quantized to extremely low bit-width, resulting in a significant performance deterioration. In contrast, our method accurately identifies the salient tokens. Additional experimental results regarding the accuracy of our method will be detailed in Section 5.2.

### 4.3 Efficient Approximation of Saliency Metric

As analyzed in Section 4.2, adaptive KV cache compression requires the explicit computation of full attention scores, as referred to Figure 4(b), which clashes with fast attention implementations like FlashAttention [8, 7, 9]. As shown in Figure 4(c), FlashAttention computes attention outputs in tiles without storing the intermediate attention scores. To reconcile the efficiency of FlashAttention with the substantial compression offered by adaptive KV caching, we devise an effective approximation for Eq. 8 as a measure of token saliency. Specifically, we sample a small group of tokens, designated

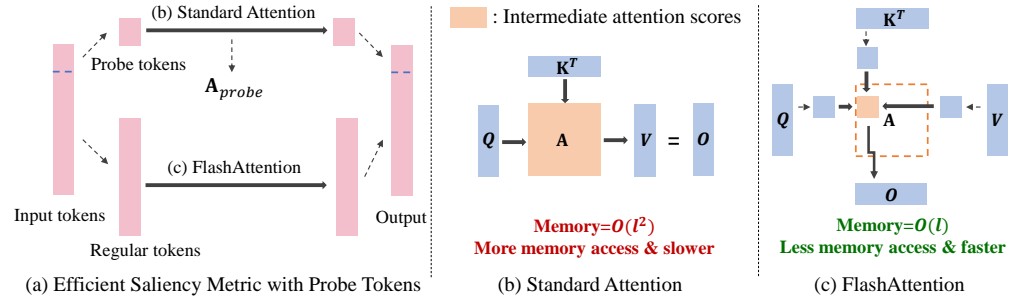

(a) Efficient Saliency Metric with Probe Tokens  (b) Standard Attention  (c) FlashAttention

Figure 4: (a): Efficient saliency metric only requires attention scores of probe tokens through standard attention, enabling fast computation for the majority of tokens through FlashAttention. (b): In standard attention, full attention scores are computed before deriving the attention output. (c): FlashAttention avoids large attention matrix memory transfers by partitioning input matrices into blocks for incremental computation.

as **probe tokens**, and compute their attention scores $\mathbf{A}_{probe}$ as follows:

$$\mathbf{A}_{probe} = \text{Softmax}\left(\frac{\mathbf{Q}_{probe}\mathbf{K}^T}{\sqrt{d_k}}\right). \tag{9}$$

By substituting $\mathbf{A}_{probe}$ into Eq. 8, we can approximate the saliency of all tokens. For the remaining non-probe tokens, their attention scores do not have to be computed explicitly, enabling the integration of fast attention implementations to expedite the generation process, as illustrated in Figure 4(a).

However, the positions of the probe tokens will undoubtedly affects the accuracy of the approximated token saliency and the selection of probe tokens is under explored. In this study, we suggest four strategies for sampling probe tokens:

- **Random tokens**. The probe tokens are randomly sampled from all positions.

- **Special tokens**. The special tokens and punctuation tokens will be treated as probe tokens.

- **Recent tokens**. The most recent tokens are selected as probe tokens.

- **Random+recent tokens**. The probe tokens will be divided into two parts, one using recent tokens and the other randomly selecting from the remaining tokens.

Table 2: Performance comparisons of various probe strategies. Data is collected from LLaMA3-8B model on GSM8k dataset. We quantize 40% salient tokens to 4-bit and the remaining 60% tokens to 2-bit. The proportion of probe tokens is 10%.

| Probe Strategy | Acc.(%) |
|---|---|
| All tokens | 52.54 |
| Random tokens | 47.46 |
| Special tokens | 46.78 |
| Recent tokens | 51.10 |
| Random+recent tokens | **52.08** |

It should be emphasized that our study diverges from prior research [16] in that, rather than directly choosing special or recent tokens as salient tokens, we opt to sample a subset of tokens as "probes" to detect the salient ones. As depicted in Table 2, we present a comprehensive comparison of the performance among four distinct sampling strategies. Among the four strategies examined, a hybrid approach that combines recent tokens with randomly selected tokens emerges as the most effective. Unless otherwise specified, this hybrid strategy with $5\%$ recent tokens and $5\%$ random tokens will be employed in our method.

## 5 Experiment

### 5.1 Implementation Details

**Models and datasets.** To validate the efficacy of our proposed method, we conduct experiments with three open-source LLMs: Mistral [20], LLaMA2 [37] and LLaMA3. These models are evaluated on three challenging benchmarks: GSM8k [6] for math problem solving, HumanEval [4] for code generation, and Line Retrieval [25] for data retrieval. To ensure reproducibility, the reported results are obtained using the Language Model Evaluation Harness [15] and LongEval [24].

**Quantization and generation settings.** We employ mixed precision quantization for KV cache where salient tokens will be quantized to 4-bit while the remaining will be quantized to 2-bit. For both subsets, we apply channelwise quantization for the key cache and channel-separable tokenwise quantization for the value cache. The proportion of salient tokens will be denoted by "Saliency Ratio" in the experimental results. During the decoding process, ZipCache adopts a streaming strategy [21] and repeats the compression process for the KV cache whenever 100 new tokens are generated.

## 5.2 Comparison with SOTA methods

### 5.2.1 Evaluation on GSM8k

We begin our evaluation on GSM8k dataset with chain-of-thoughts (CoT) prompting, and the results are presented in Table 3. This task requires LLM to solve mathematical problems and return the final answer without multiple options. This task poses considerable challenges and previous KV cache compression methods manifest notable declines in accuracy. For instance, KIVI [32] shows an accuracy drop of 7.89% on LLaMA3-8B model, indicating the suboptimality of preserving recent tokens in full precision instead of identifying salient ones. Moreover, there is a substantial decrease in accuracy, amounting to $20.4\%$, for MiKV [43] under the high compression ratio. This suggests that accumulated attention scores mistakenly identify salient tokens, resulting in the loss of vital information during compression. By contrast, the proposed normalized attention scores can accurately measure token saliency, leading to a substantial enhancement in accuracy by 18.27% for LLaMA3-8B models in comparison to MiKV. In comparison to GEAR [21], which quantizes the entire KV cache to 4-bit, our approach additionally quantizes $40\%$ tokens to 2-bit with enhanced performance on Mistral-7B model. This underscores the superiority of accurate adaptive compression of KV cache.

Table 3: Performance comparisons on GSM8k with CoT prompts. Here, "H/L" denotes the bit-width for salient tokens (high-precision) and regular tokens (low-precision), respectively. The compression ratio is calculated with an average input length of $l = 840$.

| Model | Method | Bit-width (H/L) | Saliency Ratio | Compression Ratio | Acc.(%) |
|---|---|---|---|---|---|
| Mistral-7B | FP16 | 16/16 | 100% | 1× | 41.62 |
| | H2O [46] | 16/0 | 40.0% | 2.50× | 1.67 |
| | GEAR [21] | 4/4 | 100% | 3.00× | 39.42 |
| | KIVI [32] | 16/2 | 15.2% | 3.46× | 39.04 |
| | MiKV [43] | 4/2 | 60.0% | 4.98× | 36.32 |
| | ZipCache | 4/2 | 60.0% | **4.98×** | **41.24** |
| LLaMA2-7B | FP16 | 16/16 | 100% | 1× | 14.18 |
| | H2O [46] | 16/0 | 40.0% | 2.50× | 13.50 |
| | GEAR [21] | 4/4 | 100% | 3.00× | 12.96 |
| | KIVI [32] | 16/2 | 15.2% | 3.46× | 13.19 |
| | MiKV [43] | 4/2 | 60.0% | 4.98× | 9.02 |
| | ZipCache | 4/2 | 60.0% | **4.98×** | **13.50** |
| LLaMA2-13B | FP16 | 16/16 | 100% | 1× | 28.05 |
| | H2O [46] | 16/0 | 40.0% | 2.50× | 26.00 |
| | GEAR [21] | 4/4 | 100% | 3.00× | 25.40 |
| | KIVI [32] | 16/2 | 15.2% | 3.46× | 27.29 |
| | MiKV [43] | 4/2 | 60.0% | 4.98× | 23.65 |
| | ZipCache | 4/2 | 60.0% | **4.98×** | **27.85** |
| LLaMA3-8B | FP16 | 16/16 | 100% | 1× | 55.88 |
| | H2O [46] | 16/0 | 40.0% | 2.50× | 27.82 |
| | GEAR [21] | 4/4 | 100% | 3.00× | 49.43 |
| | KIVI [32] | 16/2 | 15.2% | 3.46× | 47.99 |
| | MiKV [43] | 4/2 | 70.0% | 4.69× | 35.48 |
| | ZipCache | 4/2 | 70.0% | **4.69×** | **53.75** |

## 5.3 Evaluation on HumanEval

In this subsection, we assess the performance of code generation across various KV cache compression methods, as summarized in Table 4. Remarkably, ZipCache attains a compression ratio of $4.94\times$ without sacrificing performance when tested with the Mistral-7B model, outperforming predecessor methods. Moreover, when evaluating on LLaMA3-8B model, our approach outperforms KIVI-2 [32] by 7.32% with a significantly higher compression ratio ($4.39\times$ vs. $2.55\times$). It should be noted that the

average input length for this task is only 119, while KIVI retains the recent 32 tokens in full-precision, thereby considerably diminishing its overall compression ratio. This underscores the advantage of ZipCache over methods that consistently retain information of recent tokens.

Table 4: Performance comparisons on HumanEval for code generation. Here, "H/L" denotes the bit-width for salient tokens (high-precision) and regular tokens (low-precision), respectively. 0-bit denotes the tokens are evicted. The compression ratio is calculated with an average input length of $l = 120$.

| Model | Method | Bit-width (H/L) | Saliency Ratio | Compression Ratio | Acc.(%) |
|---|---|---|---|---|---|
| | FP16 | 16/16 | 100% | 1× | 29.27 |
| | H2O [46] | 16/0 | 40.0% | 2.50× | 14.63 |
| Mistral-7B | GEAR [21] | 4/4 | 100% | 3.00× | 28.05 |
| | KIVI [32] | 16/2 | 26.7% | 2.55× | 28.05 |
| | MiKV [43] | 4/2 | 60.0% | 4.94× | 27.44 |
| | ZipCache | 4/2 | 60.0% | **4.94×** | **29.27** |
| | FP16 | 16/16 | 100% | 1× | 14.02 |
| | H2O [46] | 16/0 | 40.0% | 2.50× | 11.59 |
| LLaMA2-7B | GEAR [21] | 4/4 | 100% | 3.00× | **13.02** |
| | KIVI [32] | 16/2 | 26.7% | 2.55× | 11.59 |
| | MiKV [43] | 4/2 | 80.0% | 4.39× | 10.37 |
| | ZipCache | 4/2 | 80.0% | **4.39×** | 12.80 |
| | FP16 | 16/16 | 100% | 1× | 33.54 |
| | H2O [46] | 16/0 | 40.0% | 2.50× | 15.85 |
| LLaMA3-8B | GEAR [21] | 4/4 | 100% | 3.00× | 28.66 |
| | KIVI [32] | 16/2 | 26.7% | 2.55× | 25.61 |
| | MiKV [43] | 4/2 | 80.0% | 4.39× | 29.88 |
| | ZipCache | 4/2 | 80.0% | **4.39×** | **32.93** |

### 5.3.1 Evaluation on Line Retrival

We further evaluate the data retrieval performance of various KV cache compression methods on Line Retrieval [25] dataset, where LLMs are required to retrieve specific content from a record of lines using a corresponding line index. The accuracy results under various number of lines are depicted in Figure 5. Notably, all quantization-based compression methods exhibit superior performance compared to the eviction-based approach H2O [46]. For eviction-based methods, information is permanently discarded upon eviction, whereas quantization introduces only minor errors while preserving the integrity of the data. Additionally, in comparison to KIVI [32], which always maintains recent caches at full precision, our approach consistently achieves better retrieval accuracy. This can be attributed to the nature of retrieval tasks, where salient tokens may appear at any position within the context, rather than being confined to the most recent caches. Moreover, when compared to MiKV [43], which employs accumulated attention scores as a saliency metric, our method yields a remarkable 42% accuracy improvement when evaluated using 200 lines on the Mistral-7b model. This substantial enhancement once more highlights the effectiveness of normalized attention scores in identifying salient tokens.

Additional experimental results on HumanEval [4] can be found in the supplementary material.

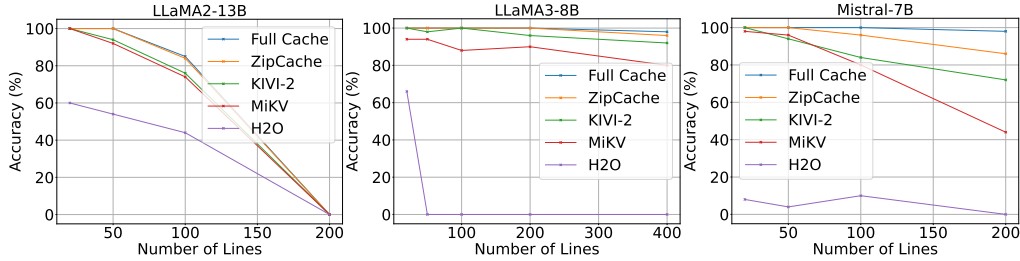

Figure 5: Performance comparisons of various KV cache compression methods on Line Retrieval.

## 5.4 Generation Efficiency

In this subsection, we compare the latency and memory consumption of ZipCache and MiKV [43] under various input lengths, as depicted in Figure 6. Data is collected by serving LLaMA3-8B model on a Nvidia A100 GPU. MiKV employs accumulated attention scores to estimate token saliency, necessitating the use of standard attention for both prefill and decoding phases. Conversely, through an efficient approximate saliency metric, ZipCache requires only the calculation of the attention matrix for $10\%$ of the tokens, while the remaining $90\%$ tokens can be computed using either FlashAttention [7] or FlashDecoding [9]. Consequently, ZipCache achieves faster inference speed and lower memory usage, boasting a $37.3\%$ reduction in prefill-phase latency, a $56.9\%$ reduction in decoding-phase latency, and a $19.8\%$ reduction in GPU memory usage when the input length scales to $4096$.

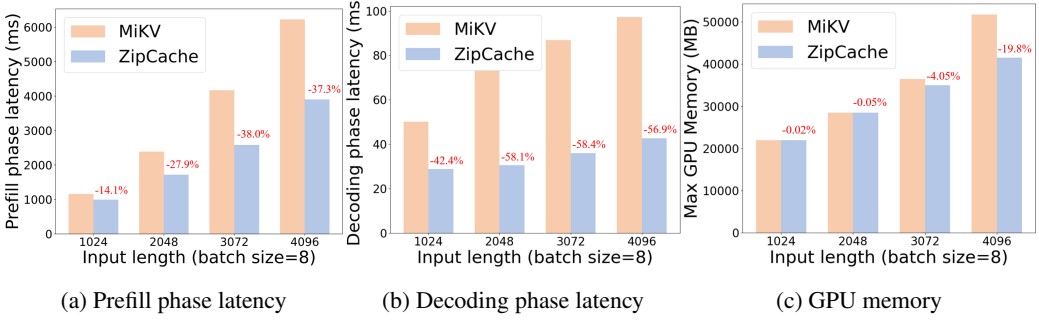

(a) Prefill phase latency      (b) Decoding phase latency      (c) GPU memory

Figure 6: Comparisons of prefill-phase, decoding-phase latency and memory consumption between MiKV and ZipCache.

## 6 Conclusion and Future Work

In this paper, we have proposed ZipCache, an accurate and efficient mixed-precision quantization framework for compressing KV cache. To commence, we introduce a channel-separable quantization scheme for KV cache, effectively reducing the overhead of storing quantization parameters compared to traditional fine-grained quantization schemes without performance degradation. Additionally, we present a novel metric for accurately assessing token saliency based on normalized attention scores. This metric enables adaptive quantization of all tokens according to their saliency, leading to improved compression ratios without sacrificing model performance. Moreover, we introduce an efficient approximation method for the token saliency metric, seamlessly integrating with fast attention implementations such as FlashAttention and FlashDecoding. This enhancement significantly boosts generation speed and reduces GPU memory requirements. Our extensive experiments have demonstrated that ZipCache achieves state-of-the-art compression performance in terms of compression ratio, accuracy and generation speed. We believe that ZipCache will pave the way for more practical and scalable deployment of LLMs in various real-world applications.

**Limitations and Broader Impacts.** While ZipCache presents promising advancements in KV cache mixed-quantization frameworks for LLMs, the saliency ratio is manually specified before evaluation and cannot be automatically adjusted based on task datasets. Moreover, similar to other generative models, ZipCache can potentially be used to generate malicious content.

**Acknowledgement** This work was supported by National Key Research and Development Program of China (2022YFC3602601).

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

# Appendix

## A    Calculation of Overhead for Different Quantization Schemes

Assuming $b = 8$, $hd = l = 4096$, and that the KV cache is quantized to 4-bit, we proceed to calculate the actual compression ratio for different quantization granularities. For groupwise quantization with a group size of $n = 32$, the compression ratio $R_{group}$ is given by:

$$R_{group} = \frac{2 \times bhld \times 16}{2 \times bhld \times 4 + \frac{4bhld}{n} \times 16} = 3.200 \tag{A}$$

For tokenwise quantization, the compression ratio $R_{token}$ can be calculated as:

$$R_{token} = \frac{2 \times bhld \times 16}{2 \times bhld \times 4 + 4 \times bl \times 16} = 3.992 \tag{B}$$

For our proposed quantization baseline, the compression ratio $R_{baseline}$ is determined by:

$$R_{baseline} = \frac{2 \times bhld \times 16}{2 \times bhld \times 4 + 3 \times hd \times 16 + 2 \times bl \times 16} = 3.995 \tag{C}$$

## B    Implementation Details of ZipCache

In this section, we provide an overview of the channel-separable tokenwise quantization scheme in Algorithm 1. Additionally, we present the process of ZipCache's prefill phase as described in Algorithm 2, as well as its decoding phase detailed in Algorithm 3. It is worth mentioning that during both the prefill and decoding phases, rather than calculating attention outputs separately for probe tokens and regular tokens followed by merging, FlashAttention [7] is utilized to compute the attention output for all tokens simultaneously. Additionally, attention scores of probe tokens are calculated. By bypassing the substantial memory accesses associated with matrix splitting and merging, this strategy enhances generation speed.

---

**Algorithm 1:** Channel-separable Tokenwise Quantization (CSTQuant)

---

**procedure** CSTQuant:
  **Input:** data $\mathbf{X} \in \mathbb{R}^{l \times hd}$, target bit-width $k$
  **for** $i \leftarrow 0$ **to** $hd$ **do**
    $\mathbf{c}_i = \sqrt{\max(|\mathbf{X}_i|)}$
    $\mathbf{X}_i = \frac{\mathbf{X}_i}{\mathbf{c}_i}$ // Normalizing each channel of $\mathbf{X}$
  $\hat{\mathbf{X}} = \text{TokenQuant}(\mathbf{X}, k)$ // Do tokenwise quantization
  **for** $i \leftarrow 0$ **to** $hd$ **do**
    $\hat{\mathbf{X}}_i = \hat{\mathbf{X}}_i \times \mathbf{c}_i$ // Rescale each channel of $\mathbf{X}$
  **return** $\hat{\mathbf{X}}$

---

## C    Additional Experimental Results

### C.1    Effect of Saliency Metric

In this section, we conduct ablation studies on our saliency metric, as shown in Table A. It demonstrates the superiority of our saliency metric over using accumulated attention scores or consistently prioritizing the latest tokens.

### C.2    Accuracy and Efficiency Comparisons of various KV cache compression methods

In this section, we present the accuracy and efficiency comparisons of various KV cache compression methods, as presented in Table B. Data is collected by evaluating LLaMA3-8B model on 200-line

**Algorithm 2:** ZipCache for Prefill Phase

---

**procedure** `ZipCachePrefill`:

    **Input:** Query states $\mathbf{Q}$, key states $\mathbf{K}$, value states $\mathbf{V}$, saliency ratio $r\%$, bit-width for salient tokens $k_h$, bit-width for regular tokens $k_l$

    `// Salient Token Identification`

    Select probe tokens and compute their attention scores $\mathbf{A}_{probe}$ by Eq. 9

    Measure the token saliency $\tilde{p}$ with $\mathbf{A}_{probe}$ by Eq. 8

    `// Computing Attention Output with FlashAttention`

    $\mathbf{O} = \text{FlashAttention}(\mathbf{Q}, \mathbf{K}, \mathbf{V})$

    `// Compressing KV Cache`

    Partition key states: $\mathbf{K}_{salient}, \mathbf{K}_{regular} = \text{Split}(\mathbf{K}, \tilde{p}, r\%)$

    Partition value states: $\mathbf{V}_{salient}, \mathbf{V}_{regular} = \text{Split}(\mathbf{V}, \tilde{p}, r\%)$

    $\mathbf{K}_{salient} = \text{ChannelQuant}(\mathbf{K}_{salient}, k_h)$, $\mathbf{V}_{salient} = \text{CSTQuant}(\mathbf{V}_{salient}, k_h)$

    $\mathbf{K}_{regular} = \text{ChannelQuant}(\mathbf{K}_{regular}, k_l)$, $\mathbf{V}_{regular} = \text{CSTQuant}(\mathbf{V}_{regular}, k_l)$

    $\hat{\mathbf{K}} = \text{Concat}(\mathbf{K}_{salient}, \mathbf{K}_{regular})$

    $\hat{\mathbf{V}} = \text{Concat}(\mathbf{V}_{salient}, \mathbf{V}_{regular})$

    `// Return Attention Output and Compressed KV Cache`

    **return O**, *($\hat{\mathbf{K}}$, $\hat{\mathbf{V}}$)*

---

**Algorithm 3:** ZipCache for Decoding Phase

---

**procedure** `ZipCacheDecoding`:

    **Input:** Query vector $\mathbf{q}$, key vector $\mathbf{k}$, value vector $\mathbf{v}$, KV cache ($\hat{\mathbf{K}}$, $\hat{\mathbf{V}}$), saliency ratio $r\%$, bit-width for salient tokens $k_h$, bit-width for regular tokens $k_l$, decoding token index $i$, probe attention score $\mathbf{A}_{probe}$

    $\mathbf{K} = \text{Concat}(\mathbf{k}, \hat{\mathbf{K}})$ `// Concatenate key cache`

    $\mathbf{V} = \text{Concat}(\mathbf{v}, \hat{\mathbf{V}})$ `// Concatenate value cache`

    $\mathbf{o} = \text{FlashAttention}(\mathbf{q}, \mathbf{K}, \mathbf{V})$ `// Compute attention output`

    $i = i + 1$

    **if** $i == 100$ **then**

        `// Re-compress every 100 tokens`

        Extract $\mathbf{K}[:-100]$ and $\mathbf{V}[:-100]$ and adaptively compress them with $\mathbf{A}_{probe}$

        Reset $i = 0$, $\mathbf{A}_{probe} = \text{None}$

    **else if** $i > 95$ **or** $randint(0, 100) < 5$ **then**

        `// probe tokens consists of 5% recent and 5% random tokens.`

        Compute attention scores $\mathbf{a}$ of current token by Eq. 4

        $\mathbf{A}_{probe} = \text{Concat}(\mathbf{a}, \mathbf{A}_{probe})$

    `// Return Attention Output, KV Cache and Attention Scores from Probe Tokens`

    **return o**, $(\mathbf{K}, \mathbf{V})$, $\mathbf{A}_{probe}$

---

retrieval task with a Nvidia A100 GPU. We use a batch size of 8 and an average input length of 3072. To ensure a fair comparison, we implement GEAR and KIVI with FlashAttention integration.

Notably, the latency of ZipCache is lower than that of GEAR, which can be attributed to our efficient quantization scheme, whereas GEAR has a high overhead due to outlier extraction. Compared to KIVI, ZipCache's latency is slightly higher (2584.01 ms vs. 2482.26 ms), but ZipCache achieves a higher compression ratio and better performance. This difference is due to KIVI's fixed compression strategy, while we adaptively compress the KV cache based on the saliency. In comparison to MiKV [43], which identifies salient tokens through accumulated attention scores, our method achieves a notable $10.0\%$ accuracy improvement by accurately pinpointing salient tokens and a substantial $38.0\%$ decrease in prefill latency by integrating FlashAttention [7].

Moreover, to demonstrate the efficacy of the proposed efficient quantization scheme, we also implement ZipCache with groupwise quantization. The results show that using groupwise quantization

Table A: The effect of various saliency metric on GSM8k with CoT prompts. Here, "H/L" denotes the bit-width for salient tokens (high-precision) and regular tokens (low-precision), respectively. "Locality" means the recent tokens are identified as salient tokens. The compression ratio is calculated with an average input length of $l = 840$.

| Model | Metric | Bit-width (H/L) | Saliency Ratio | Compression Ratio | Acc.(%) |
|---|---|---|---|---|---|
| Mistral-7B | FP16 | 16/16 | 100% | 1× | 41.62 |
| | Locality | 4/2 | 60.0% | 4.98× | 25.40 |
| | Accumulated Attention Scores | 4/2 | 60.0% | 4.98× | 38.20 |
| | Normalized Attention Scores | 4/2 | 60.0% | 4.98× | **41.24** |

increases inference speed (2664.05 ms vs. 2584.01 ms) and reduces the compression rate (3.81× vs. 4.43×) due to massive quantization overhead.

Table B: Accuracy and efficiency comparisons over LLaMA3-8B on the 200-line retrieval task. Here, "H/L" denotes the bit-width for salient tokens (high-precision) and regular tokens (low-precision), respectively. 0-bit denotes the tokens are evicted. Saliency ratio denotes the proportion of salient tokens. The compression ratio is calculated with an average input length of $l = 3072$.

| Method | Bit-width (H/L) | Saliency Ratio | Compression Ratio | Acc.(%) | Prefill-phase Latency (ms) |
|---|---|---|---|---|---|
| FP16 | 16/16 | 100% | 1× | 100 | 2340.11 |
| H2O | 16/0 | 40.0% | 2.50× | 0 | 4335.01 |
| GEAR | 4/4 | 100% | 3.00× | 100 | 2968.43 |
| KIVI | 16/2 | 8.33% | 4.36× | 96 | 2482.26 |
| MiKV | 4/2 | 80.0% | 4.43× | 90 | 4170.61 |
| ZipCache | 4/2 | 80.0% | 4.43× | 100 | 2584.01 |
| ZipCache (Groupwise Quantization) | 4/2 | 80.0% | 3.81× | 100 | 2664.05 |

## C.3 Evaluation on LongBench

In this subsection, we evaluate the performance of ZipCache on LongBench using the longchat-7b-v1.5-32k model, as shown in Table C. The results show that ZipCache outperforms the previous state-of-the-art method, KIVI on long context scenario.

Table C: Performance comparisons on LongBench.

| Model | Method | Qasper | QMSum | MultiNews | TREC | TriviaQA | SAMSum | LCC | RepoBench-P |
|---|---|---|---|---|---|---|---|---|---|
| Llama-2-7b-chat | FP16 | 21.92 | 21.01 | 26.09 | 64.00 | 83.51 | 41.5 | 58.39 | 52.26 |
| | KIVI-2 | 14.31 | **20.76** | 25.75 | 64.00 | **83.38** | 39.14 | 56.17 | 50.12 |
| | ZipCache | **20.93** | 20.69 | **26.12** | **64.50** | 83.38 | **40.11** | **56.60** | **52.00** |

