# OpenReview forum: "ZipCache: Accurate and Efficient KV Cache Quantization with Salient Token Identification"
_NeurIPS.cc/2024/Conference — NeurIPS 2024 poster_

### Official Review · Reviewer_E63Z · 2024-07-08

**Soundness:** 3
**Presentation:** 3
**Contribution:** 3
**Rating:** 7
**Confidence:** 4

**Summary:**

The paper presents an adaptive, mixed-precision quantization method for compressing KV cache in LLMs. It proposes a channel-separable tokenwise quantization scheme to establish a robust quantization baseline, reducing the memory overhead of quantization parameters. A saliency metric, based on normalized attention scores, is employed to identify salient tokens accurately. This approach preserves essential information while aggressively quantizing less salient data, allowing for higher compression ratios with minimal impact on model accuracy. The method can be integrated with FlashAttention with an efficient approximation of the proposed metric. Experimental results validate the superiority of ZipCache on speed and accuracy over existing methods.

**Strengths:**

1. The idea of preserving salient tokens while compressing less critical tokens is both intuitive and effective, with the proposed accurate saliency metric being crucial for achieving this.
2. The paper presents a novel quantization scheme tailored for channel-wise outliers in LLMs, providing a notable reduction in memory overhead compared to group-wise quantization schemes.
3. The paper demonstrates strong experimental results on GSM8k and HumanEval, convincingly showcasing the benefits of adaptive KV cache compression.
4. The integration with FlashAttention greatly enhances overall generation speed.
5. The paper is well-organized.

**Weaknesses:**

1. Evaluating ZipCache on other generation or comprehending tasks will make the findings more robust.

**Questions:**

1. Have you considered using only the attention scores from the most recent token to determine token saliency? What impact might this have on the overall performance?
2. What does saliency mean in Figure 3c? What is the probability of each token being selected as a salient token, as mentioned in the caption of Figure 3?

**Limitations:**

Yes

---

> ### Author Rebuttal · Authors · 2024-08-06
>
> Thanks to the reviewer for the valuable comments.
>
> **Q1:  Evaluating ZipCache on other generation or comprehending tasks.**
>
> As shown in Table C in the rebuttal PDF, we evaluate the performance of ZipCache on LongBench. The results show that ZipCache outperforms the previous state-of-the-art method, KIVI. Due to the limited time slot of the rebuttal, we will conduct experiments on more models and add the results to the revised version.
>
> **Q2: Using only the most recent token to determine token saliency.**
>
> As shown below, determining token saliency with only the most recent token leads to a 3.2% accuracy drop compared to ZipCache.
>
>  Table: The effect of different saliency metric on GSM8k with CoT prompts. Here, "H/L" denotes the bit-width for salient tokens (high-precision) and regular tokens (low-precision), respectively. The compression ratio is calculated with an average input length of $l=840$.
>
>  | Model       | Metric                     | Bit-width (H/L) | Saliency Ratio | Compression Ratio | Acc. (%) |
>  |-------------|----------------------------|-----------------|----------------|-------------------|----------|
>  | Mistral-7B  | FP16                       | 16/16           | 100%           | 1$\times$         | 41.62    |
>  | Mistral-7B  | Last Token's Attention Scores | 4/2             | 60.0%          | 4.98$\times$      | 38.04    |
>  | Mistral-7B  | Normalized Attention Scores | 4/2             | 60.0%          | 4.98$\times$      | **41.24** |
>
> **Q3: What does saliency mean in Figure 3c?**
>
> There are many attention layers and heads in a model. For each token, this value is derived by counting the number of times this token is selected as a salient token among all attention heads, then normalizing this count by the total number of attention heads.

---

> > ### Comment · Reviewer_E63Z · 2024-08-09
> > **Review for rebuttal**
> >
> > Thanks for the efforts from the authors. My concerns are addressed in the rebuttal. I think this paper is well-motivated and has its merit to the community. It demonstrates strong performance in the experiments. Therefore, I raise my score after rebuttal.

---

> > > ### Author Response · Authors · 2024-08-10
> > > **Appreciation for Your Valuable Feedback**
> > >
> > > Dear Reviewer E63Z,
> > >
> > > Thank you for your feedback. We truly appreciate your careful consideration of our responses.
> > >
> > > Best regards,
> > >
> > > Authors of #5969.

---

### Official Review · Reviewer_S3gk · 2024-07-11

**Soundness:** 3
**Presentation:** 3
**Contribution:** 4
**Rating:** 8
**Confidence:** 5

**Summary:**

The paper introduces ZipCache, an adaptive KV cache compression method for LLMs by accurately identifying salient tokens. It first presents a channel-separable tokenwise quantization scheme that reduces the overhead of quantization parameters. Next, it proposes a metric for identifying salient tokens based on normalized attention scores. By efficiently approximating the saliency metric, the method integrates seamlessly with fast attention mechanisms like FlashAttention. The efficacy of ZipCache is demonstrated through extensive experiments, showing superior performance in terms of compression ratio, speed, and minimal accuracy loss compared to existing methods.

**Strengths:**

1.The introduction of normalized attention scores as a metric for token saliency is significant and promising for adaptive KV cache compression. It can accurately identify salient tokens, marking a substantial improvement over prior methods.

2.The approximated saliency metric can be integrated with FlashAttention, making it practical for real-world applications.

3.The idea of the channel-separable tokenwise quantization approach is novel and well-motivated, reducing memory overhead compared to groupwise counterparts.

4.The experimental results are promising, clearly demonstrating the efficacy of ZipCache in terms of compression ratio, speed, and minimal accuracy loss.

5.The paper is clearly written and easy to follow.

**Weaknesses:**

1.The experiments should encompass a broader diversity of tasks to comprehensively assess the method's effectiveness, such as LongBench.

2.The font size in Figures 5 and 6 is too small, making them difficult to read. It is recommended to increase the font size for better clarity and accessibility.

**Questions:**

See weaknesses

**Limitations:**

See weaknesses

---

> ### Author Rebuttal · Authors · 2024-08-06
>
> Thanks to the reviewer for the valuable comments.
>
> **Q1: Evaluate ZipCache over LongBench.**
>
> As shown in Table C in the rebuttal PDF, we evaluate the performance of ZipCache on LongBench. The results show that ZipCache outperforms the previous state-of-the-art method, KIVI. Due to the limited time slot of the rebuttal, we will conduct experiments on more models and add the results to the revised version.
>
> **Q2: The font size in Figures 5 and 6 is too small.**
>
> Thanks for your valuable comment. We will revise it in the final version.

---

> > ### Comment · Reviewer_S3gk · 2024-08-12
> >
> > Thank the authors' detailed response. After reviewing all comments and responses, my concerns are addressed well. The proposed method is novel and effective, and I believe this paper is ready to be published. So I would like to raise my score to 8.

---

> > > ### Author Response · Authors · 2024-08-13
> > > **Appreciation for Your Valuable Feedback**
> > >
> > > Dear Reviewer S3gk,
> > >
> > > Thank you for your feedback. We truly appreciate your careful consideration of our responses.
> > >
> > > Best regards,
> > >
> > > Authors of #5969.

---

### Official Review · Reviewer_Nc8R · 2024-07-16

**Soundness:** 3
**Presentation:** 3
**Contribution:** 2
**Rating:** 5
**Confidence:** 4

**Summary:**

This paper proposes a post training quantization framework named ZipCache for quantizing the key and value cache of LLMs. The authors introduce a channel-separable token-wise quantization scheme, which consumes less memory than the group quantization in terms of the quantization parameters. They also select saliency tokens based on a normalized attention matrix to avoid the accumulated bias. Additionally, the authors introduce an approximation method to apply it to FlashAttention.

**Strengths:**

- Overall, the writing of the paper is clear and easy to follow. Especially, the figures are drawn well.
- By proposing the ZipCache, this work promotes the PTQ quality of LLMs on three chanllenging datasets.
- The methods are well-integrated into popular framwork like FlashAttention.

**Weaknesses:**

- The idea of channel-separable quantization is not novel. It is a common practice to smooth the outliers of activations in the channel dimension, e.g., SmoothQuant [1], OmniQuant [2].
- As shown in Table 3 (a), the values on the diagonal of the matrix are relatively large, while the values at other positions are relatively small. Based on this circumstance,  if the mean operation is performed for none-zero positions, earlier tokens may have smaller $p_i$ for they have more tiny values counted. Therefore, it may cause a prefer to the latest tokens. I think this is a issue for the proposed technique in Section 4.2, which needs more analysis.
- Ablation studies are not provided.
- As stated in Line 268 in the paper, new tokens remain un-quantized unless they reach 100. As far as I know, keeping part of tokens in full precision can improve the accuracy [3, 4]. Therefore, it is better to distinguish the contributions of this implementation and the proposed techniques.
- I think the value of 'H/L' should be carefully considered. This is because both KIVI and ZipCache keep new tokens in FP16 at first and then perform quantization. Since the 'H' value of KIVI is 16, why it becomes 4 for ZipCache?
- The equation (5) is not accurate for not subtracting the zero-point 'z' in the de-quantization.

[1]. SmoothQuant: Accurate and Efficient Post-Training Quantization for Large Language Models.

[2]. Omniquant: Omnidirectionally calibrated quantization for large language models.

[3]. KIVI: ATuning-Free Asymmetric 2bit Quantization for KV Cache.

[4]. SKVQ: Sliding-window Key and Value Cache Quantization for Large Language Models.

**Questions:**

- The authors have compared the channel-separable quantization with many dynamic methods, but lacks comparison of accuracy with static methods, e.g., WKVQuant [1]. It is better to further explore whether the $c_i$ in equation (6) can outperform well-trained parameters, since they consumes the same memory space.
- How does the calculation of $c_i$ in equation (6) effect the inference latency? It is better to compare the channel-separable quantization with group quantization and static methods in terms of the inference latency.

[1]. WKVQuant: Quantizing weight and key/value cache for large language models gains more.

**Limitations:**

This paper contains the describtion of limitations. For suggestions, please refer to the Weakness and Questions.

---

> ### Author Rebuttal · Authors · 2024-08-06
>
> Thanks to the reviewer for the valuable comments.
>
> **Q1: The idea of channel-separable quantization is not novel.**
>
> Please refer to General Response Q1.
>
> **Q2: Performing the mean operation may cause a prefer to the latest tokens.**
>
> Indeed, there might be a prefer to the latest tokens since the values on the diagonal of the matrix are relatively large. However, the overall token saliency is adaptively determined in our method. As shown in Figure A in the rebuttal PDF, we present the token saliency with a 100-line retrieval sample as input. Besides the latest tokens, the system prompts in the front and related context in the middle are also assigned high saliency by our method. Moreover, Table B in the rebuttal PDF shows that our method achieves superior performance compared to using accumulated attention scores or a fixed prefer to the latest tokens.
>
> **Q3: Ablation studies are not provided.**
>
> As referred to Table 1 and Table 2 in the paper, we have conducted ablation experiments to demonstrate the effectiveness of channel-separable quantization scheme and the efficient approximation of the saliency metric, respectively. We further conduct ablation studies on our saliency metric, as shown in Table B in the rebuttal PDF. It demonstrates the superiority of our saliency metric over using accumulated attention scores or consistently prioritizing the latest tokens.
>
> **Q4: Remaining new tokens un-quantized can improve accuracy.**
>
> During the prefill phase, KIVI [i] maintains full-precision KV caches for recent tokens to ensure accuracy, while our approach **adaptively quantizes all KV caches**. During the decoding phase, we quantize all KV caches with mixed-precision every 100 tokens, while KIVI adopts a sliding window and always keeps KV caches for latest tokens (128 tokens by default) in full-precision. This implies that KIVI always retains more KV cache at full precision compared to our method. Moreover, the streaming strategy we adopted aligns with GEAR [ii] and **is aimed at enhancing decoding speed**. Therefore, the comparison with KIVI and GEAR fairly demonstrates the efficacy of our method.
>
> **Q5: The value of `H/L' should be carefully considered.**
>
> As referred to Q4, during the prefill phase, we quantize all KV caches, with the bit-width for salient tokens set to 4. This approach contrasts with KIVI's method of maintaining full-precision (FP16) for the recent tokens.
>
> **Q6: The equation (5) is not accurate.**
>
> Thanks for your valuable comment. Equation (5) has been revised as follows:
>
> $\hat{\mathbf{x}}=\mathcal{Q}_U(\mathbf{x},k)=(\mathrm{clip}(\lfloor \frac{\mathbf{x}}{s}\rceil +z, 0, 2^{k}-1) - z) \cdot s.$
>
>
> **Q7: Comparisons between channel-separable quantization and static methods like WKVQuant.**
>
> Due to different settings, ZipCache and WKVQuant [iii] are not directly comparable. WKVQuant quantizes model weights and KV cache together, optimizing parameters through cross-block reconstruction regularization and a gradient descent algorithm. In contrast, our method focuses solely on KV cache compression and is entirely training-free.
>
> **Q8: Compare the channel-separable quantization with group quantization and static methods in terms of the inference latency.**
>
> Inference latency comparisons among channel-separable quantization, groupwise quantization, and static quantization methods are detailed in Table A of the rebuttal PDF. Using static quantization can slightly reduce latency (2556.03 ms vs. 2584.01 ms). Conversely, groupwise quantization increases inference speed (2664.05 ms vs. 2584.01 ms) and reduces the compression rate (3.81$\times$ vs. 4.43$\times$) due to massive quantization overhead.
>
> [i] KIVI: A Tuning-Free Asymmetric 2bit Quantization for KV Cache.
>
> [ii] GEAR: An Efficient KV Cache Compression Recipe for Near-Lossless Generative Inference of LLM.
>
> [iii] WKVQuant: Quantizing Weight and Key/Value Cache for Large Language Models Gains More.

---

> > ### Comment · Reviewer_Nc8R · 2024-08-13
> > **Review for Rebuttal**
> >
> > Thanks to the authors for the detailed response. I understand the motivation of the channel-seperable token-wise quantization, as well as the effectiveness of the token saliency metric after the rebutall. The integration of this kind of methods to flash-attention is also a valuable contribution. Therefore, considering the answers from Q1 to Q5, I decide to raise my score to 5. Although, I think the description of the Section 4.1 can be improved, especially from the aspect of accuracy and latency. As claimed to be a more advanced baseline, it should process better generalization ability compared to group quantization and well-trained static parameters, rather than solely performing well on a single dataset. Additionally, I think the answers to Q6 and Q8 should be included into the main text.

---

> ### Author Response · Authors · 2024-08-10
> **Follow-Up on Rebuttal**
>
> Dear Reviewer Nc8R,
>
> We greatly appreciate the time and effort in reviewing our work. We have carefully considered your comments and suggestions and made significant revisions to address the concerns you raised. We are eager to ensure that our paper meets the high standards of our respected reviewers.
>
> Please don’t hesitate to let us know if there is any additional feedback you might have at this stage.
>
> Best regards,
>
> Authors of #5969.

---

> ### Author Response · Authors · 2024-08-13
> **Follow-Up on Rebuttal**
>
> Dear Reviewer Nc8R,
>
> Thank you for dedicating your time to reviewing our paper. As the discussion period deadline is approaching, we kindly invite any further comments or concerns you might have. Your feedback has been immensely valuable to us for refining the paper.
>
> Best regards,
>
> Authors of #5969.

---

> ### Author Response · Authors · 2024-08-13
> **Appreciation for Your Valuable Feedback**
>
> Dear Reviewer Nc8R,
>
> Thank you for your feedback. We truly appreciate your careful consideration of our responses and will carefully revise the paper based on your and other reviewers' suggestions.
>
> Best regards,
>
> Authors of #5969.

---

### Official Review · Reviewer_ozSB · 2024-07-17

**Soundness:** 3
**Presentation:** 2
**Contribution:** 2
**Rating:** 5
**Confidence:** 5

**Summary:**

The paper introduces a channel-separable quantization scheme that decouples the quantization along channel and token dimensions. This method significantly reduces the quantization overhead without compromising performance. To accurately recognize salient tokens, the paper introduces a new token saliency metric based on normalized attention scores, which alleviates the bias towards earlier tokens that accumulate more values. The authors have demonstrated on different LLM models with different tasks.

**Strengths:**

The idea of channel separable token saliency quantization seems to work better than standard groupwise quantization.

The results are strong and beats the SoTA!

The demonstration with flashattention integration is useful

**Weaknesses:**

The paper needs to be proof read by a native English speaker, to improve its readability.

The contribution is incremental, and the idea of group quantization (different for different K and V) are not new as already noted by the authors.

The idea of speeding up through flashattention and flashdecoding while the baselines not leveraging that is a bit of unfair comparison. Both GEAR and KIVI can exploit similar speed up benefit, thus Fig 1 comparison and later in table speed comparison is not fair.

The details of system level implementation is missing, it will be good to help the reviewer provide how the speed up can be demonstrated on a single GPU inference system.

**Questions:**

refer to weakness.

**Limitations:**

The system benefits are not clear or thorough for the reviewer to clearly appreciate. Also more fair system eval is needed.

---

> ### Author Rebuttal · Authors · 2024-08-06
>
> Thanks to the reviewer for the valuable comments.
>
> **Q1: The writing can be more native.**
>
> Thanks for your valuable comment. We will carefully proofread the paper in the final version.
>
> **Q2: The idea of group quantization (different for different K and V) is not new.**
>
> There might be some misunderstandings regarding groupwise quantization. Groupwise quantization is a scheme built upon tokenwise quantization, further processing outlier channels in distinct groups, as illustrated in Figure 2(c) of the paper. Additionally, applying different quantization schemes for K and V respectively (channelwise quantization for K and channel-separable tokenwise quantization for V) is not the key contribution of our paper. In terms of the quantization scheme, our main contribution lies in the introduction of channel-separable quantization to the KV cache compression, significantly reducing the overhead of quantization parameters without sacrificing performance, as detailed in General Response Q1.
>
> **Q3: The contribution is incremental.**
>
> Please refer to General Response Q1 for the detailed contribution of our channel-separable quantization scheme and to General Response Q2 for an elaboration on the overall contribution of our paper.
>
> **Q4: The speed comparison is not fair.**
>
> Thank you for your valuable comments. Firstly, we highlight that we are the pioneering work that enables adaptive KV cache compression to be compatible with FlashAttention, greatly enhancing the generation speed compared to previous approaches like H2O [i] and MiKV [ii]. Moreover, the reported speed results of GEAR [iii] and KIVI [iv] were obtained with their official implementations, which did not integrate FlashAttention at the time of our paper submission. To ensure a fair comparison, we implement GEAR and KIVI with FlashAttention integration, and the results are shown in Table A in the rebuttal PDF. Notably, the latency of ZipCache is lower than that of GEAR, which can be attributed to our efficient quantization scheme, whereas GEAR has a high overhead due to outlier extraction. Compared to KIVI, ZipCache's latency is slightly higher (2584.01 ms vs. 2482.26 ms), but ZipCache achieves a higher compression ratio and better performance. This difference is due to KIVI's fixed compression strategy, while we adaptively compress the KV cache based on the saliency. These results will be revised in the final version.
>
>
> **Q5: The details of system level implementation are missing.**
>
> The detailed processes of ZipCache for both prefill and decoding phases are summarized in Algorithms 2 and 3 in the Appendix of our paper. Currently, we implement ZipCache based on the Hugging Face Transformers library, with specific modifications to the KV cache module. Our method is also orthogonal to other system level frameworks such as vLLM [v]. It should be noted that we utilize FlashAttention to maximize computational efficiency for both the prefill and decoding phases, eliminating the need to customize additional matrix multiplication kernels. We will release the source code upon acceptance.
>
> [i] H2O: Heavy-Hitter Oracle for Efficient Generative Inference of Large Language Models.
>
> [ii] No Token Left Behind: Reliable KV Cache Compression via Importance-Aware Mixed Precision Quantization.
>
> [iii] GEAR: An Efficient KV Cache Compression Recipe for Near-Lossless Generative Inference of LLM.
>
> [iv] KIVI: A Tuning-Free Asymmetric 2bit Quantization for KV Cache.
>
> [v] Efficient Memory Management for Large Language Model Serving with PagedAttention.

---

> > ### Comment · Reviewer_ozSB · 2024-08-13
> > **Response.**
> >
> > Thanks to the authors for the new results with attached pdf. Particularly, I appreciate the fair pre-fill stage comparison with all quantization schemes leveraging flashattention.
> >
> > 1. While I appreciate the author's effort, I think further clarifications are needed. In specific, the KV compression can be useful for longer context based evaluation, the current manuscript including that in the  one in rebuttal pdf. The LLaMA2 model has pretraining context length of ~4k which is not sufficient enough for the long context demonstration. Beyond long context evaluation the benefits of KV cache compression is not that usefully demonstrable. And the main manuscript demonstrates results with around 900 and 200 tokens on average as said by the authors.
> >
> > 2. The idea of normalized attention score is not new either, so I believe their is a bit of overclaim here. Please refer to this paper, that proposed something similar, namely MAS [1].
> >
> > 3. Why the generation efficiency is compared with only MiKV, instead of other SoTA quantization methods like KIVI? It is understandable that MiKV falls short as it cant leverage flash-decoing/attention, this is again unfair comparison.
> >
> > 4. It is also not clear if the probe tokens are selected randomly how it will still help manage perform flashattention or flashdecoding efficiently. Also, during decode phase do you implement flashattention or flashdecoding? please provide more details.
> >
> > 5. Overall, I believe the paper is built on top of the main findings of probe tokens, however the details in that front is yet to be sufficient. Additionally, how the selection of probe tokens help guide the bit width assignment of salient tokens if a token does not have any representative in the randomly selected probe token set?
> >
> > 6. The algorithm of decode stage seems ambiguous, or not fully informative. For example, the new tokens getting added on top of previous KV, is it the hybrid H/L quantized KV or or some high precision KV? As we may need to compute attention and softmax etc at high precision to have numerical stability. This needs clarification.
> >
> >
> > 7. It looks like the probe tokens during decode phase are kept at higher precision, did the author include that in their total compression scheme? Also, the probe token computation and memory overhead is not clearly discussed in the paper's results. This should impact accuracy as well as throughput compared to schemes like KIVI.
> >
> >
> >
> > [1] On the Efficacy of Eviction Policy for Key-Value Constrained Generative Language Model Inference, 2024.

---

> ### Author Response · Authors · 2024-08-10
> **Follow-Up on Rebuttal**
>
> Dear Reviewer ozSB,
>
> We greatly appreciate the time and effort in reviewing our work. We have carefully considered your comments and suggestions and made significant revisions to address the concerns you raised. We are eager to ensure that our paper meets the high standards of our respected reviewers.
>
> Please don’t hesitate to let us know if there is any additional feedback you might have at this stage.
>
> Best regards,
>
> Authors of #5969.

---

> ### Author Response · Authors · 2024-08-13
> **Follow-Up on Rebuttal**
>
> Dear Reviewer ozSB,
>
> Thank you for dedicating your time to reviewing our paper. As the discussion period deadline is approaching, we kindly invite any further comments or concerns you might have. Your feedback has been immensely valuable to us for refining the paper.
>
> Best regards,
>
> Authors of #5969.

---

> ### Author Response · Authors · 2024-08-13
> **Response to additional questions (Part 1)**
>
> Thanks to the reviewer for the valuable comments.
>
> **Q1: The KV compression can be useful for longer context based evaluation.**
> Firstly, in alignment with prior literature [1-4], we have evaluated our method on three challenging and widely-recognized benchmarks, including the long context Line Retrieval task. As highlighted in Table 3, our approach consistently surpasses previous state-of-the-art methods, underscoring its effectiveness. Secondly, as mentioned in lines 36-38 of our paper, during **batch inference**, the KV cache with an input length of approximately 4K is already a significant bottleneck in the storage system. For example, the KV cache can occupy 1.2TB of memory space with a batch size of 64 and an input length of 4096. Moreover, to further address your point, we have also evaluated the performance of ZipCache on LongBench using the longchat-7b-v1.5-32k model, as shown below. Due to the limited time slot of the rebuttal, we will conduct experiments on more long-context tasks and add the results to the revised version.
>
> Table: Performance comparisons on LongBench with LongChat-v1.5-7B-32k. The saliency ratio is 60% for ZipCache.
> | Model                  | Method  | TREC↑  | SAMSum↑ | LCC↑   | RepoBench-P↑ |
> |------------------------|---------|-------|--------|-------|-------------|
> | **LongChat-v1.5-7B-32k**| FP16    | 66.06 | 41.19  | 52.96 | 56.8        |
> |                        | KIVI-2  | **66.0** | 40.57 | 47.99 | 52.6      |
> |                        | ZipCache| **66.0** | **40.64**  | **51.25** | **52.87** |
>
> **Q2: Similar idea of normalized attention scores has been proposed.**
> Thank you for sharing this literature [5] and we will include it in the references in the revised version.
> However, we would like to emphasize that our proposed saliency metric can be applied universally to all tokens without exception, whereas the approach in [5] excludes certain tokens from the eviction scope based on their standard deviation. Additionally, our work provides a comprehensive analysis of the limitations of using accumulated attention scores as a saliency metric, as discussed in lines 200-210 and illustrated in Figure 3 of the paper.
>
> **Q3: The generation efficiency is compared with only MiKV and the comparison is unfair.**
> As referred to Figure 1 in the paper, Table A in the Appendix and Table A in the rebuttal PDF, we have compared the generation efficiency with H2O, GEAR and KIVI. Moreover, the previous adaptive KV cache compression methods [1-2] are not compatible with FlashAttention, which is a major drawback of them. By contrast, our method integrates seamlessly with FlashAttention, enhancing generation speed.
>
> **Q4: How do probe tokens help perform flashattention or flashdecoding efficiently?**
> Firstly, as referred to lines 230-232 and Equation (9) of the paper, we select a small set of probe tokens and compute attention scores using their queries, which allows us to approximate the saliency for all tokens. Once we have determined the token saliency, we can efficiently compute the attention output using FlashAttention. For a more detailed explanation of our method, please refer to Algorithm 2 and Algorithm 3 in the Appendix.
>
> **Q5: Do you implement flashattention or flashdecoding during decoding phase?**
> Yes. As mentioned in lines 267-268, 304-307, and detailed in Algorithm 3 in the Appendix, we implement FlashDecoding during the decoding phase. Similar to the prefill phase, we explicitly compute attention scores for a small set of probe tokens to approximate the saliency of all tokens, enabling the majority of tokens to be computed using FlashDecoding.
>
> **Q6: How does the selection of probe tokens help guide the bit width assignment of salient tokens if a token does not have any representative in the probe token set.**
> As mentioned in lines 254-255 of the paper, our approach to selecting probe tokens involves a **hybrid** strategy, where 5% of the tokens are the **latest** ones, and 5% are randomly selected. All previous tokens are attended when computing attention scores for the latest token.
>
> **Q7: The algorithm of decode stage seems ambiguous. For example, what is the precision of the KV cache for new tokens.**
> As referred to lines 267-268 of the paper, we implement a streaming strategy during the decoding phase, where the new KV cache is quantized every 100 new tokens generated. This approach is consistent with the strategy used in GEAR [3] and is designed to enhance decoding speed. Consequently, before reaching the 100-token threshold, the new KV cache is maintained in full precision. After 100 new tokens are generated, all KV cache will be quantized based on their estimated saliency.
>
> **Q8: Attention and softmax need to be computed at high precision to have numerical stability.**
> Indeed. The KV cache will be dequantized to full-precision when calculating attention. This is consistent with previous KV cache quantization work [2-4].

---

> ### Author Response · Authors · 2024-08-13
> **Response to additional questions (Part 2)**
>
> **Q9: The probe tokens during decoding phase are kept at higher precision.**
> This is not the case. All KV caches, including those for probe tokens, are quantized after every 100 new tokens are generated. As referred to Q5, during the decoding phase, the primary difference between probe tokens and other tokens is that we use the queries of probe tokens to compute attention scores explicitly, while other tokens are processed using FlashDecoding.
>
> **Q10: Keeping probe tokens at higher precision will impact accuracy as well as throughput.**
> Please refer to Q9. Probe tokens are not kept in higher precision.
>
> **Q11: What is the computation and memory overhead for probe tokens?**
> As shown below, computing attention scores with the queries of probe tokens introduces limited computation and memory overhead.
>
> Table: Computation and memory overhead for probe tokens. Here, "ZipCache w/o Probe Tokens" denotes the token saliency is randomly generated rather than approximated with probe tokens. Data is collected by serving LLaMA3-8B on a NVIDIA A100 GPU with a batch size of 8 and sequence length of 3072.
>
> | Model                  | Method  | Prefill-phase Latency (ms)  | Max GPU Memory (MB)|
> |------------------------|---------|-------|--------|
> | **LLaMA3-8B**          | ZipCache w/o Probe Tokens| 2503.06 | 34990 |
> |                        | ZipCache w/ Probe Tokens| 2584.01 | 34992  |
>
>
> [1] H2O: Heavy-Hitter Oracle for Efficient Generative Inference of Large Language Models.
>
> [2] No Token Left Behind: Reliable KV Cache Compression via Importance-Aware Mixed Precision Quantization.
>
> [3] GEAR: An Efficient KV Cache Compression Recipe for Near-Lossless Generative Inference of LLM.
>
> [4] KIVI: A Tuning-Free Asymmetric 2bit Quantization for KV Cache.
>
> [5] On the Efficacy of Eviction Policy for Key-Value Constrained Generative Language Model Inference.

---

> > ### Comment · Reviewer_ozSB · 2024-08-13
> > **Final remark**
> >
> > Thanks to authors for their comprehensive response! Overall majority of my concerns seems to be addressed by the authors. However, regarding Q3, I would recommend the authors to change their wordings as it is not true that majority of the quantization can not support flashattention. Flashattention is an orthgonal method that can be merged with most of the quantization schemes. Please tone down this claim about others don't supporting it. For Q4, if so, then please explicitly mention in L307 of the manuscript that flashattention during prefill and decoding during decode, despite this being intuitive (as flashattn does not provide much benefit during decode phase). I thus increase my score.

---

> > > ### Author Response · Authors · 2024-08-14
> > > **Appreciation for Your Valuable Feedback**
> > >
> > > Dear Reviewer ozSB,
> > >
> > > Thank you for your feedback. We truly appreciate your careful consideration of our responses and will carefully revise the paper based on your and other reviewers' suggestions.
> > >
> > > Best regards,
> > >
> > > Authors of #5969.

---

### Author Rebuttal · Authors · 2024-08-06

We thank all reviewers for their valuable feedback. Overall, our work has been well recognized as
- "It is novel and well-motivated" (Reviewers S3gk and E63Z)
- "It is well-integrated into FlashAttention" (All Reviewers)
- "It is clear and easy to follow" (Reviewers Nc8R, S3gk, and E63Z)
- "It demonstrates strong performance" (Reviewers ozSB, S3gk, and E63Z).

We have summarized and addressed the main concerns as follows:

**Q1: The proposed channel-separable quantization is not novel.**

We highlight a significant challenge in KV cache groupwise quantization: the number of quantization parameters grows linearly with the product of sequence length and hidden dimension, as referred to Table 1 in the paper, which greatly impacts the KV cache compression ratio. This crucial issue is overlooked in previous literature and motivates us to disentangle quantization along channel and token dimensions to reduce quantization overhead. Therefore, **our motivation is fundamentally different from model quantization such as SmoothQuant [i] and OmniQuant [ii]**, which focus on migrating quantization difficulties between activations/weights or query/key states before matrix multiplications. As referred to Table 1 in the paper, our scheme significantly reduces the overhead of quantization parameters and achieves superior performance compared to groupwise quantization.

**Q2: The contribution of our paper.**

Our contributions are summarized as follows:
1) We propose a channel-separable quantization scheme to reduce the overhead of quantization parameters. It brings a higher compression ratio and achieves superior performance compared to groupwise quantization.
2) We introduce an accurate metric to identify salient tokens and adaptively quantize all KV caches based on their saliency, thereby improving the overall compression ratio.
3) ZipCache is the pioneering work that enables adaptive KV cache compression to be compatible with FlashAttention, significantly enhancing the generation speed and practicality. Overall, as referred to Table 3 in the paper, ZipCache achieves the **highest KV cache compression ratio** and the **highest accuracy** compared to predecessor methods, demonstrating the efficacy and contribution of our method.

**Q3: Additional experiments on LongBench.**

As shown in Table C in the rebuttal PDF, we evaluate the performance of ZipCache on LongBench. The results show that ZipCache outperforms the previous state-of-the-art method, KIVI [iii]. Due to the limited time slot of the rebuttal, we will conduct experiments on more models and add the results to the revised version.

[i] SmoothQuant: Accurate and Efficient Post-Training Quantization for Large Language Models.

[ii] Omniquant: Omnidirectionally Calibrated Quantization for Large Language Models.

[iii] KIVI: A Tuning-Free Asymmetric 2bit Quantization for KV Cache.

---

### Decision · Program_Chairs · 2024-09-25

**Decision:**

Accept (poster)

**Comment:**

This paper presents ZipCache for KV cache quantization with salient token identification. After the rebuttal, it receives one strong accept, one accept, and two borderline accept. Its merits, including clear motivation, good organization, interesting idea, and good results, are well recognized by the reviewers. The authors' response well resolves the concerns of unsatisfying writing, insufficient implementation details, and incorrect formulation. Based on the enclosed reviews, I think this work should be accepted. Please also incorporate these details in the revised manuscript.